# Recent Updates on Risk and Management Plans Associated with Polypharmacy in Older Population

**DOI:** 10.3390/geriatrics7050097

**Published:** 2022-09-13

**Authors:** Asim Muhammed Alshanberi

**Affiliations:** 1Department of Community Medicine and Pilgrims Health Care, Umm Alqura University, Makkah 24382, Saudi Arabia; amshanberi@uqu.edu.sa or asim.alshanberi@bmc.edu.sa; Tel.: +966-555533389; 2School of Medicine, Batterjee Medical College for Sciences and Technology, Jeddah 21442, Saudi Arabia

**Keywords:** family medicine, intervention, management, outcome, polypharmacy

## Abstract

The concept of polypharmacy encompasses adverse drug reactions and non-adherence factors in elderly individuals. It also leads to the increased use of healthcare services and negative health outcomes. The problem is further alleviated by the odds of potentially inappropriate medications (PIM), which lead to the development of drug-related problems. Since polypharmacy is more commonly observed in the elderly population, urgency is required to introduce operative protocols for preventing and managing this problem. The family medicine model of care can be associated with favorable illness outcomes regarding satisfaction with consultation, treatment adherence, self-management behaviors, adherence to medical advice, and healthcare utilization. Hence, interventions built on family medicine models can provide significant support in improving the outcomes of the older population and their quality of life. In this regard, the authors have taken up the task of explaining the accessible resources which can be availed to improve the application of health care services in the field of geriatric medicine.

## 1. Introduction

Polypharmacy is the term given when multiple drugs are prescribed to treat diseases and improve other health conditions [1,2,3]. The WHO defines polypharmacy as the concurrent use of five or more medications by a patient which may include over-the-counter, prescription, and/or traditional and complementary medicines [4]. This process represents a major concern for older adults. This population suffers from multiple chronic conditions, e.g., depression, arthritis, asthma, coronary heart disease, diabetes, hypertension, vision and hearing impairment, lower lean body mass, mobility, and chronic obstructive pulmonary disease [5,6,7,8]. Major adverse outcomes associated with polypharmacy includes increased length of stay in hospital, mortality, falls and readmission to hospital soon after discharge [9,10]. Moreover, the probability of these undesirable effects rises when the medications increase in number. Hence, inappropriate polypharmacy (excessive or unnecessary usage of several medications) elevates the risk of adverse drug effects which can lead to drug–disease interactions and cognitive impairment, which worsens another disease or eventually leads to the occurrence of new one [11].

The complications of polypharmacy can be reduced with the intervention of a professionally trained health care team dedicated to the care of geriatrics. In the context of Saudi Arabia, a continuous growth in the life expectancy of the general population has been observed in the last decade. The elderly population, accounting for 2.96% in 2011 to 3.65% in 2021, are projected to constitute 25% of the population by 2050. However, the proportion of people aged 80 years and above may reach 4% by the same year, and this population is considered to be growing faster, as reflected in the significant growth noticed in the healthcare and management sector of Saudi Arabia. On the other hand, it raises drug related issues, e.g., adverse drug reaction, adverse drug events, and drug interactions [4]. Most of the identified geriatric cases are prescribed with anticholinergics, sedatives, and hypnotics, cardiovascular agents, analgesics, and antidiabetic agents, which carry potential risk in the case of polypharmacy. Such cases can be reviewed with the assigned tools. If needed, a de-prescribing protocol must be developed and implemented in selected cases to avoid complications of adverse drug reactions, avoid high risk medications, and reduce the inapplicable medications, if prescribed [12].

Continuous educational activities are required that include education of the elderly population in relation to the medication intake, summarizing the strategies to reduce polypharmacy, and development of inter-professional team relations and attitude to reduce the effects of polypharmacy to a degree of 40% in the case of untoward reactions [4], regarding which budding healthcare professionals may lack knowledge and encounter difficulties. Hence, in this regard, the authors have taken up the task of compiling the modern methods of trouble shooting the issues related to the polypharmacy so that the information can be used by the readers such that better health care can be provided in relation to the management of polypharmacy related issues in the case of geriatrics.

## 2. Complications Leading to Polypharmacy

The risk factors for polypharmacy can occur either (or both) at the level of the health care system and patient [13,14]. For example, maintenance of poor medical record leads to polypharmacy when such ceased medications are automatically prescribed by a physician. However, at the patient level, polypharmacy is mainly recognized in the elderly group as they are more vulnerable to suffer from one or more chronic conditions which subsequently leaves them with a long list of medication. Moreover, the elderly population with no primary care physician or multiple subspecialist physicians are also at great risk of polypharmacy [15,16].

Polypharmacy in the young population occurs due to heart diseases, diabetes, cancer, stroke, chronic pain such as fibromyalgia, developmental disabilities, and chronic medical disorders as they involve multiple treatments and modalities [17]. Polypharmacy becomes more severe in the case of mentally challenged candidates. Such individuals stay on psychotropic medications with adverse effects and numerous medications are prescribed to reduce their side effects. Patients suffering from chronic neurological disorders and similar medical complications also need multiple medications for preventing the progress of disease [18]. The occurrence of adverse drug events (ADEs) is associated with increased morbidity and mortality, prolonged hospitalizations, and higher cost of care. Moreover, Stevenson and coworkers have recently observed the association of frailty with medication-related harm in a prospective observational cohort study through an integrated health and social care approach, to reduce inappropriate polypharmacy [19].

## 3. Mode of Assessment for Potentially Inappropriate Medication

Several assessment tools have been used earlier (Figure 1) to trace the potentially inappropriate medication (PIM). They include the Medication Appropriateness Index (MAI), Beers, screening tool of older people’s prescriptions (STOPP), and screening tool to alert for right treatment (START). Explicit tools, such as STOPP and START, assist with easy and quick decision-making criteria. These tools help in comparing patient’s medication list with PIM which leads to disease interactions, medication and medication duplication [20,21]. Beers criteria further classify PIM by disease state and class of drug [22]. START and STOPP scale are applied in combination for identifying medications which are considered unsuitable (STOPP) followed by the application of alternative medications for treating the disease (START) [20]. MAI represents an implicit assessment tool as it considers patient complexity. It is more time-consuming and patient-centered, involving physician judgment in contrast to the set guidelines of the assessment tools [23]. The attitude, experience, and knowledge of a physician limit this assessment method inherently, and hence it is less reliable than explicit mode of assessment in providing meaningful insight to a clinical problem. MAI poses ten questions, including consideration of medication duplications; medication requirement; medication and disease interactions; dosage suitability, drug formulation and treatment duration; and directions for use, and optimal therapy for diseases and conditions. It should be noted that MAI takes appreciable time to prescribe medication even though these questions seem clear and straightforward [24,25,26].

It should be noted that none of the PIMs tools exhibit complete efficacy in improving patient-related outcomes. Hence, these procedures can lead to unnecessary polypharmacy risks [27]. One of the prominent strategies used to reduce pill burden involves deprescribing unnecessary medications by evaluating the patient’s active medication lists. It helps in reducing the negative impact of ADE and financial hardship. Hence, deprescribing offered an initial therapeutic intervention plan and management protocol, simultaneously with the introduction of clinically appropriate therapy [28,29,30]. Point-of-care tools also help the patient to understand initially the urgency to cut down the unnecessary medication, thereby reducing the risks of polypharmacy. This step leads to priorities and preferences required to implement the prescription of new drugs for slowing down the progression of disease and to address the symptoms for improving the health of an individual [31,32].

## 4. Deprescribing Approaches

Deprescribing is a patient-centered process of medication withdrawal intended to achieve improved health outcomes through discontinuation of one or more medications that are either potentially harmful or no longer required [33]. It is a five-step, patient-centered systematic approach (Figure 2) used for evaluating and discontinuing medications in patients where potential harms exceed potential benefits in relation to individual existing functioning capability, care goals, preferences, values and life expectancy. Hence, this process suspends medication, changes medication, and decreases medication dosages for obtaining the best clinical outcomes [34]. This is because deprescribing approaches are patient-specific and focused and follow interventions with appreciable variability in medications used and patient characteristics [35]. The process of deprescription should be initiated as a “therapeutic intervention” simultaneously by introducing clinically appropriate therapy. Hence, it becomes imperative for a physician to examine patient approach on goals of therapy. It also takes into consideration the basic parameters of chronic conditions and medications and affords prime concern regarding prescription to slow down the disease progression along with minimizing the health decline [36,37]. It has been observed that only a limited percentage of the elderly community debate health-related decision-making priorities with their primary care physicians. Hence, in order to optimize disease control, the physician should examine specific therapeutic goals at every patient visit. Additionally, health systems and practices need to adopt rationalized and updated methods for tracking and medication reconciliation as up-to-date medication lists form the basis of identifying potential medications for deprescribing, which could minimize the burden of staff, patient, and physician [38,39,40]. Sawan et al. (2020) have recently summarized the common deprescribing interventions, including enablers and barriers to implement deprescribing across settings (e.g., primary, secondary, residential care facilities) and current deprescribing polices in place internationally [41]. Several deprescribing tools have been used recently, e.g., mobilizing community pharmacists [42], creating evidence-based deprescribing guidelines [43], geriatric pharmacoeconomics [44], and engaging primary care providers in deprescribing trials [45,46]. The dedicated team involved in the management of health-related issues with the elderly patients shall be advised to follow the process of deprescribing medication qualitatively (focused groups) and quantitatively by Delphi’s criteria [12]. Further, Bruyere deprescribing guidelines could prove beneficial in case of managing the deprescription in relation to the elderly patients in case of antihypertensive medications, psychotic medications, proton pump inhibitors, antihistamines, anti-hyperglycemic agents, sedatives and hypnotics, anticholinergics, and cholinesterase inhibitors; and OncPal deprescribing guidelines in the case of oncological and palliative care medications [47].

## 5. General Medications That Cause Problems in Elderly Population

Polypharmacy is an area of concern for elderly because such individuals are at a greater risk for adverse drug reactions (ADR) which lead to metabolic changes and reduced drug clearance [48,49,50]. This risk is furthermore exacerbated by the increasing number of drugs used. Polypharmacy may sometimes lead to “prescribing cascades.” A prescribing cascade arises when signs and symptoms (multiple and nonspecific) of an ADR are misinterpreted as a disease and a new treatment/drug therapy is further added to the earlier prescribed treatment to treat the condition. This increases the potential to develop further side-effects, leading to a prescribing cascade [51]. It has been proposed earlier by several groups of researchers that the use of multiple medications comes with an increased risk for negative health outcomes, such as higher healthcare costs, ADEs, drug–drug interactions, medication non-adherence, decreased functional status, and geriatric syndromes [52,53,54,55]. The medications listed in Table 1 cause problems in elderly individuals, when prescribed in combination or even alone.

## 6. Management Plan

Appropriate polypharmacy deals with the prescription of multiple medications in an optimized way to an individual suffering from complex conditions or for multiple conditions according to best evidence. The management plan for polypharmacy involves medication assessment and home assessment. It also incorporates balance, gait, and strength assessment (Figure 3). These are achieved partly or as a whole by eliminating duplicate medications, conducting medication reconciliations at care transition, assessing drug–drug interactions, and by reviewing drug dosages to reduce the incidence of polypharmacy [70,71,72,73,74]. It also ensures patient safety and reduces hospitalization, thereby decreasing the associated costs. It has been well documented that various tools and treatment approaches are used in parallel as there exist no ideal tools to manage polypharmacy in the elderly population [75]. It depends on the application of medications and time suitability which enables the users to employ one of existing interventions and/or tools to obtain the optimized results. Several researchers concluded that the utilization of a drug based upon the explicit assessment tools offered the best practical approach [76,77,78,79,80,81].

The World Health Organization has identified five sets of factors that affect patients’ adherence to therapy. They include socio-economic, health care team-, and system-, condition-, therapy-, and patient-related factors. These factors including health care team and system-related factors are mostly dependent on patient characteristics and experiences. For example, a good patient–provider relationship can improve adherence [82]. However, in particular, patients’ satisfaction with therapy has been linked to adherence [83,84]. Cognitive impairment in older adults has a variety of possible causes, including medication side effects; metabolic and/or endocrine derangements; delirium due to illness (such as a urinary tract or related infection); depression; and dementia, with Alzheimer’s dementia being most common. Some causes, e.g., medication side effects and depression, can be reversed or improved with treatment. Others, such as Alzheimer’s, cannot be reversed, but symptoms can be treated for a period of time, and importantly, families can be prepared for predictable changes and address safety concerns [85].

It has been documented that a multidisciplinary team involving nurses may facilitate future treatment and rehabilitation in geriatric patients with cardiovascular disorders [86]. Dedicated centralized team leaders are best suited in implementing the guidelines for residential aged care facility residents requiring specialized geriatric care via telehealth and counselling [87]. It has also been observed that the enthusiastic involvement of pharmacists in deprescription has proved beneficial in the case of elderly patients in all types of resource settings [88]. The medication management goals followed by the suggested guidelines may ensure that older patients may have a realistic understanding of their medical conditions. This pre-intervention workup, combined with engagement with family members and interdisciplinary teams, may improve post-interventional outcomes [89]. Additionally, artificial intelligence applications and computational tools could serve as a potential tool in developing management plans for individualized geriatric management of healthy aging [90].

Comprehensive geriatric assessment (CGA) involves a multidisciplinary diagnostic and treatment process for identifying medical, psychosocial, and functional limitations of a frail older person in order to develop a coordinated plan to maximize overall health with aging [91]. CGA assessment tools can be in the form of a pre-visit questionnaire which can serve as a timesaving method to gather a large amount of information. Major components of CGA that should be evaluated include functional capacity, fall risk, cognition, mood, polypharmacy, nutrition/weight change, urinary incontinence, vision/hearing, dentition, living situation, social support, financial concerns, goals of care, spirituality, and advance care preferences. Incorporating CGA for evaluating the functional outcomes in transition care using a suite of assessment tools was feasible and enabled a holistic assessment [92,93,94].

## 7. Conclusions

Here, it has been overviewed that polypharmacy is linked with duplicated therapy and contraindicated drug combinations. The symptoms caused by polypharmacy vary from tiredness, sleepiness, decreased alertness, constipation, diarrhea or incontinence, loss of appetite, confusion, falls, depression or lack of interest in usual activities, to weakness, tremors, visual or auditory hallucinations, anxiety or excitability, and/or dizziness. In this regard, communication among elderly individuals and physicians needs to be improved in applying a proper explicit or implicit tool to minimize adverse consequences of polypharmacy by reducing the pill burden. The process of deprescription should be initiated as a therapeutic intervention by introducing clinically appropriate therapy. Additionally, health systems need to adopt rationalized and updated methods for tracking and medication reconciliation as up-to-date medication lists form the basis of identifying potential medications for deprescribing.

## Figures and Tables

**Figure 1 geriatrics-07-00097-f001:**
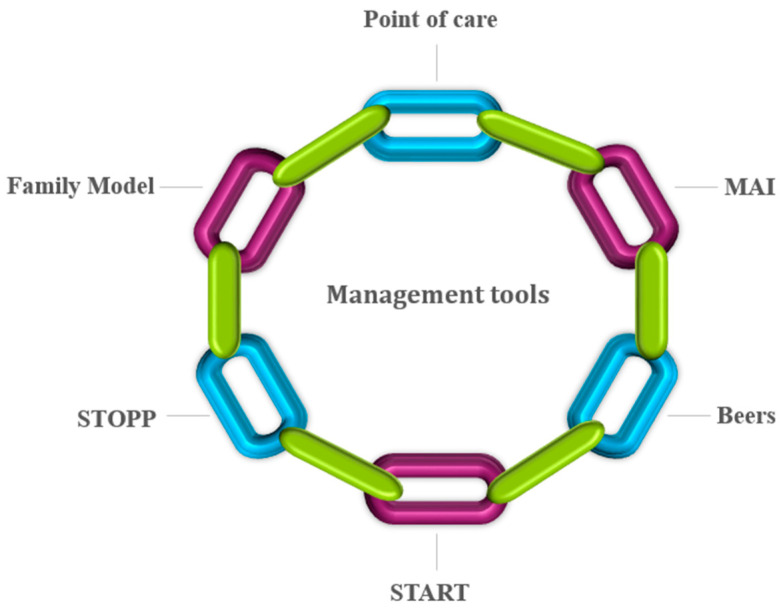
Assessment tools for tracing potentially inappropriate medications.

**Figure 2 geriatrics-07-00097-f002:**
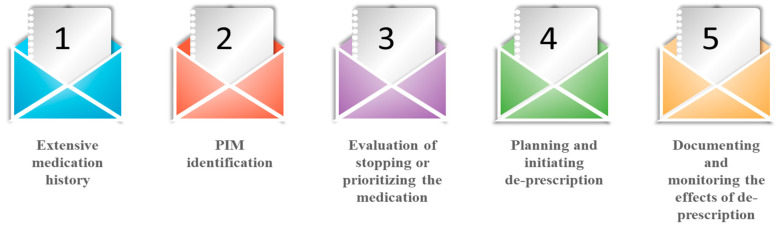
Deprescription process.

**Figure 3 geriatrics-07-00097-f003:**
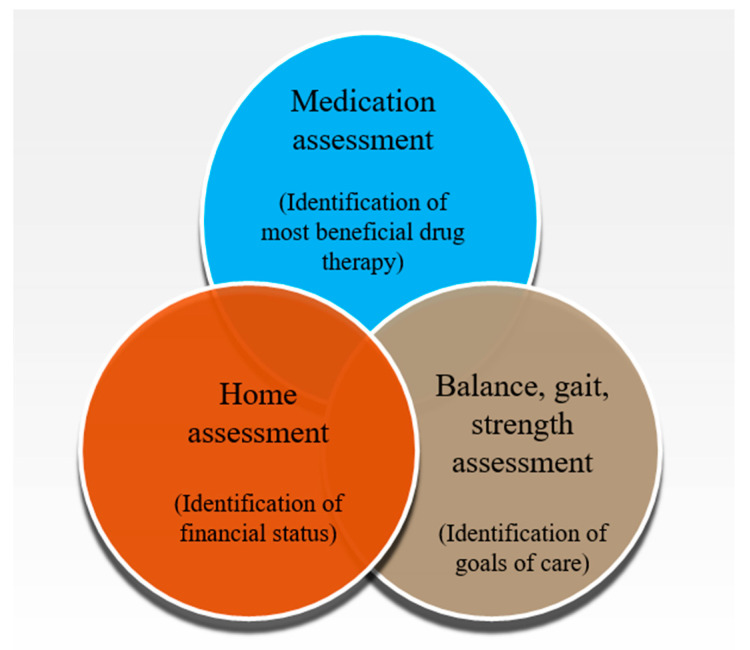
Management plans for polypharmacy.

**Table 1 geriatrics-07-00097-t001:** Adverse drug events of some medications in older population.

S. No.	Class of Drug	Medication	ADE
1	Antipsychotic	Chlorpromazine	Ischemic stroke [56]
2	Antidepressant	Doxepin	Somnolence and headache [57]
3	Anticonvulsants	Carbamazepine	Drug-drug interaction [58]
4	Glycosides	Digoxin	Confusion and nausea [59]
5	NSAIDs	Indomethacin	Behavioral problems, gastrointestinal and renal adverse effects [60]
6	Antimanic	Lithium carbonate	Transient thyrotropin elevation [61]
7	Opioid analgesics	Meperidine, propoxyphene	Hallucination [62]
8	Antihypertensives	Methyldopa	Dizziness [63]
9	Muscle relaxant	Methocarbamol	Falls and fractures [64]
10	Antiarrhythmic	Procainamide	Orthostatic hypotension [65]
11	Anticoagulants	Warfarin	Thromboembolic events [66]
12	Alkaloids	Reserpine	Deplete catecholamines [67]
13	Xanthine	Theophylline	Nausea, Loss of apetite [68]
14	Antipyretic	Paracetamol	Liver injury [69]

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
