# Peer review of "Recent Updates on Risk and Management Plans Associated with Polypharmacy in Older Population"

_geriatrics, 2022, doi:10.3390/geriatrics7050097_

Round 1

Reviewer 1 Report

This manuscript lacks the depth of information, clear objectives, and presents repetitive information that is outdated.

The author might be a novice to manuscript conceptualization and writing. For example, the brief introduction is a copy of the abstract. Furthermore, the author failed to state where the deprescribing process occurs? In other words, in which country does the author try to explain these steps? For example, what is the relevance of the deprescribing process for a reader from the US or Europe? How is the legislation differs in these countries?

The manuscript has major grammar issues and needs support. For example, “This process can elevate adverse drug events (ADEs) risk eventually.”

The author failed to provide references throughout the manuscript. For example, the author mentions Beers and other tools; however, they are not referenced. In addition, there are paragraphs without a reference. This is unacceptable.

On page 3, the author presents a figure without any reference. It is assumed the author developed the figure. If not, please reference it since it is not your work. The lack of references is a major failure of this manuscript.

The major failure of the table is that some of the medications presented are not in usage in the US. Furthermore, the information presented in the table is not novel. On the contrary, it is common knowledge for a pharmacist and repeats outdated information. Furthermore, the language is inappropriate for a publication because it is misleading. "The medications listed in table 1 causes severe problems in elderly individuals, when prescribed in combination or even alone." This sentence is misleading because it states, "severe problems", without any examples. These patients suffer from chronic conditions, and the therapy management is complex and needs careful monitoring.

Another major concern of this manuscript is that the table lists medications by brand and generic names interchangeably. It also provides the same medication twice (Acetaminophen and Paracetamol). 

The conclusion is vague and does not relate to the rest of the article.

Author Response

The author appreciates the critical comments raised by the esteemed reviewers. Efforts were raised to revise the manuscript as suggested. The changes have been marked in red color in the revised manuscript.

Reviewer # 1

  1. This manuscript lacks the depth of information, clear objectives, and presents repetitive information that is outdated. The author might be a novice to manuscript conceptualization and writing. For example, the brief introduction is a copy of the abstract. Furthermore, the author failed to state where the deprescribing process occurs? In other words, in which country does the author try to explain these steps? For example, what is the relevance of the deprescribing process for a reader from the US or Europe? How is the legislation differs in these countries?

The manuscript containing studies on deprescription from various countries has been cited in the revised manuscript (reference no 39).

  1. The manuscript has major grammar issues and needs support. For example, “This process can elevate adverse drug events (ADEs) risk eventually.”

The sentence has been edited (Page 3). As suggested, the English of the entire manuscript has been edited by a native English speaker.

  1. The author failed to provide references throughout the manuscript. For example, the author mentions Beers and other tools; however, they are not referenced. In addition, there are paragraphs without a reference. This is unacceptable.

The references are provided as suggested (Page 4).

  1. On page 3, the author presents a figure without any reference. It is assumed the author developed the figure. If not, please reference it since it is not your work. The lack of references is a major failure of this manuscript.

All the Figures are conceived by the author. They are not copied from any source.

  1. The major failure of the table is that some of the medications presented are not in usage in the US. Furthermore, the information presented in the table is not novel. On the contrary, it is common knowledge for a pharmacist and repeats outdated information. Furthermore, the language is inappropriate for a publication because it is misleading. "The medications listed in table 1 causes severe problems in elderly individuals, when prescribed in combination or even alone." This sentence is misleading because it states, "severe problems", without any examples. These patients suffer from chronic conditions, and the therapy management is complex and needs careful monitoring.

This basic information contain in the Table 1 is required for the readers, hence it is mentioned in the text.

The word severe is removed to avoid confusion.

  1. Another major concern of this manuscript is that the table lists medications by brand and generic names interchangeably. It also provides the same medication twice (Acetaminophen and Paracetamol). 

Repetition is deleted from the Table as suggested by the esteemed reviewer.

  1. The conclusion is vague and does not relate to the rest of the article.

Conclusion is revised now.

Reviewer 2 Report

The article is nicely written and will be a nice addition to the literature as an overall primer on the topic.  

Suggest adding a little about the intersection between adherence and poly pharmacy and medication related harm:

Stevenson et al. A multi-centre cohort study on healthcare use due to medication-related harm: the role of frailty and polypharmacy. Age and ageing. 2022 Mar;51(3):afac054.

Suggest editing that de-prescribing is referred to as "de-prescribing".

Suggest using a definition of deprescribing the first time it is used.  One of the most common is "patient-centred process of medication withdrawal intended to achieve improved health outcomes through discontinuation of one or more medications that are either potentially harmful or no longer required":

Page et al. A concept analysis of deprescribing medications in older people. Journal of Pharmacy Practice and Research. 2018 Apr;48(2):132-48.

Suggest pointing out that the PIMs tools are different in each country:

Lee et al. Applicability of explicit potentially inappropriate medication lists to the Australian context: A systematic review. Australasian Journal on Ageing. 2022 Jan 13.

Author Response

The author appreciates the critical comments raised by the esteemed reviewers. Efforts were raised to revise the manuscript as suggested. The changes have been marked in red color in the revised manuscript.

Reviewer # 2

The article is nicely written and will be a nice addition to the literature as an overall primer on the topic.  

  1. Suggest adding a little about the intersection between adherence and poly pharmacy and medication related harm: Stevenson et al. A multi-centre cohort study on healthcare use due to medication-related harm: the role of frailty and polypharmacy. Age and ageing. 2022 Mar;51(3):afac054.

Added in the revised manuscript (Reference no 17).

  1. Suggest editing that de-prescribing is referred to as "de-prescribing".

De-prescribing has been mentioned as "deprescribing" throughout the text.

  1. Suggest using a definition of deprescribing the first time it is used.  One of the most common is "patient-centred process of medication withdrawal intended to achieve improved health outcomes through discontinuation of one or more medications that are either potentially harmful or no longer required": Page et al. A concept analysis of deprescribing medications in older people. Journal of Pharmacy Practice and Research. 2018 Apr;48(2):132-48.

 Added on page 5 as suggested in the revised manuscript (Reference no 31).

  1. Suggest pointing out that the PIMs tools are different in each country: Lee et al. Applicability of explicit potentially inappropriate medication lists to the Australian context: A systematic review. Australasian Journal on Ageing. 2022 Jan 13.

Added as suggested in the revised manuscript (Reference no 30).

Reviewer 3 Report

Thanks for recommending me as a reviewer. In this perspective study, the authors conducted a study on the topic of recent updates on risk and management plans in older population. Patients suffering from chronic neurological disorders and similar medical complications also need multiple medications for preventing the progress of disease. This process can elevate adverse drug events risk eventually. The research is overall well written. If the authors summarized the conclusion section more clearly, it may help the reader's understanding.

Author Response

The author appreciates the critical comments raised by the esteemed reviewers. Efforts were raised to revise the manuscript as suggested. The changes have been marked in red color in the revised manuscript.

Reviewer # 3

Thanks for recommending me as a reviewer. In this perspective study, the authors conducted a study on the topic of recent updates on risk and management plans in older population. Patients suffering from chronic neurological disorders and similar medical complications also need multiple medications for preventing the progress of disease. This process can elevate adverse drug events risk eventually. The research is overall well written. If the authors summarized the conclusion section more clearly, it may help the reader's understanding.

Conclusion has been modified in the revised manuscript (Conclusion section, page 9).

Round 2

Reviewer 1 Report

The introduction is simply a rewording of the abstract and does not expand far enough on the ideas being presented. There are also numerous grammatical errors throughout the entire draft. These errors alone make the article unpublishable and require correction. The authors also provide several figures throughout the article but do not clarify if they created the figures or if they adapted them from other authors. This is a critical point that needs to be addressed. There are many grammatical errors throughout the manuscript. For example, “Point of care tools also helps in deprescribing…” This should read as “help”. This occurs multiple times throughout the article. Readers are still unsure of where this information becomes applicable. The author mentions Australia in the introduction but is a part of a university in Saudi Arabia so the reader is uncertain where the author is trying to make an impact since it is not clear.

The table presented on page 4 of the article does not present any novel information. Clinical pharmacists worldwide are already aware of these potential drug issues with these drug classes. In addition, some of the medications presented in the table are not even of use in the US so they are irrelevant to practice here in the US. The table is just listed as medications “causing problems” in elderly population. This title lacks useful information and does not add anything to current literature.

The title on page 4 of “General medications that cause problems in elderly population” is vague and doesn’t present anything novel to the literature. The paragraph states, “Potential of drug-drug interactions is further increased by use of multiple drugs.” This is something that every single pharmacist is already aware of and does not add anything new to the literature.

The conclusion does not provide any limitations or any comment to where future work is needed. The second sentence states that “symptoms caused by polypharmacy is unfortunately usually demented with the normal aging signs and symptoms…”. This sentence uses inappropriate verbiage since many elderly patients suffer with dementia, the verb “demented” is inappropriate and does not read well. Another example of a grammatical error in the conclusion rests in the last sentence where the author states, “…as up to date medication lists forms the basis…”. This should read “form the basis...”. There are multiple errors like this throughout the paper and make it difficult for the reader to understand the point.

Author Response

The author appreciates the critical comments raised by the esteemed reviewer as well as the academic editor. Efforts were raised to revise the manuscript as suggested. The changes have been marked in red color in the revised manuscript.

Reviewer

  • The introduction has been updated. English of the MS has been re-checked by a native English speaker
  • All figures have been conceptualized and prepared by the author using the templates available at https://www.slideegg.com/.
  • Grammatical errors have been minutely observed and corrected throughout the manuscript. Moreover, one of the grammatical errors as suggested by the esteemed reviewer “Point of care tools also helps in deprescribing…” has been edited.
  • The reference mentioning details about Australian population was suggested by another reviewer. However, as suggested, it has been removed in the revised MS.
  • The table has been updated as per the suggestion of the Academic Editor.
  • One of the paragraphs stating “Potential of drug-drug interactions is further increased by use of multiple drugs” has been removed in the revised MS.
  • The sentence “symptoms caused by polypharmacy are unfortunately usually demented with the normal aging signs and symptoms…” has been removed to avoid confusion.
  • Efforts were raised to remove the grammatical error in the revised MS.